# Characterising the Response of Human Breast Cancer Cells to Polyamine Modulation

**DOI:** 10.3390/biom11050743

**Published:** 2021-05-17

**Authors:** Oluwaseun Akinyele, Heather M. Wallace

**Affiliations:** Institute of Medical Sciences, School of Medicine, Medical Sciences and Nutrition, University of Aberdeen, Aberdeen AB25 2ZD, UK; seunomotayo67@gmail.com

**Keywords:** polyamines, cell growth, breast cancer subtypes, DFMO

## Abstract

Breast cancer is a complex heterogeneous disease with multiple underlying causes. The polyamines putrescine, spermidine, and spermine are polycationic molecules essential for cell proliferation. Their biosynthesis is upregulated in breast cancer and they contribute to disease progression. While elevated polyamines are linked to breast cancer cell proliferation, there is little evidence to suggest breast cancer cells of different hormone receptor status are equally dependent on polyamines. In this study, we characterized the responses of two breast cancer cells, ER+ (oestrogen receptor positive) MCF-7 and ER- MDA-MB-231 cell lines, to polyamine modulation and determined the requirement of each polyamine for cancer cell growth. The cells were exposed to DFMO (a polyamine pathway inhibitor) at various concentrations under different conditions, after which several growth parameters were determined. Exposure of both cell lines to DFMO induced differential growth responses, MCF-7 cells showed greater sensitivity to polyamine pathway inhibition at various DFMO concentrations than the MDA-MB-231 cells. Analysis of intracellular DFMO after withdrawal from growth medium showed residual DFMO in the cells with concomitant decreases in polyamine content, ODC protein level, and cell growth. Addition of exogenous polyamines reversed the cell growth inhibition, and this growth recovery appears to be partly dependent on the spermidine content of the cell. Similarly, DFMO exposure inhibits the global translation state of the cells, with spermidine addition reversing the inhibition of translation in the breast cancer cells. Taken together, these data suggest that breast cancer cells are differentially sensitive to the antitumour effects of polyamine depletion, thus, targeting polyamine metabolism might be therapeutically beneficial in breast cancer management based on their subtype.

## 1. Introduction

Breast cancer is a malignant epithelial tumour of tissues of the breast including the inner layer of the milk glands or lobules, the milk ducts, and the connective tissues [1]. It is the second most diagnosed cancer and the fifth leading cause of cancer-related death. Breast cancer is a complex heterogeneous disease with multiple underlying causes. Dysregulated metabolic/signalling pathways are major drivers of cancer [2,3,4]. Among the metabolic pathways that have been shown to be enhanced by various cellular and molecular processes that lead to breast carcinogenesis is the polyamine metabolic pathway [5,6,7,8].

The polyamines are ubiquitous polycationic molecules essential for normal cell functioning and are elevated in vast majority of human cancers [9,10,11,12]. In breast cancer, the polyamine biosynthetic pathway is upregulated, causing a 3- to 6-fold increase in tumour polyamine content compared to the normal epithelial breast tissue [13,14]. These increases in polyamines and polyamines metabolic enzymes are associated with poor prognosis of breast and colorectal cancers [15,16,17]. Similarly, many indicators of poor prognosis in breast cancer, such as tumour size and histological grade are correlated with increased tumour polyamine content [14], highlighting the importance of polyamines in breast carcinogenesis and disease progression. While elevated polyamines are associated with poor prognosis in breast cancer, it is not clear if breast cancer cells of different hormone receptor status show similar dependence on polyamines for continued proliferation.

The aim of this study was to characterize the growth responses of two breast cancer cells of different subtypes (based on their hormone receptor status) to conditions that altered their polyamine content, and to determine the sensitivity of the breast cancer cells to the anti-tumour effect of polyamine pathway inhibition. To inhibit the polyamine pathway, a well-known inhibitor (α-difluoromethylornithine, DFMO) of a key enzyme (ornithine decarboxylase, ODC) in polyamine biosynthesis was used. As part of our study, we developed a novel assay to quantify the intracellular DFMO content after cell exposure. We also determined the requirement of each polyamine of both breast cancer cell lines for growth and studied the effect of intracellular polyamine modulation on the protein translation state of the cells.

## 2. Materials and Methods

### 2.1. Materials

The following were purchased from Sigma-Aldrich (Dorset, UK): Dulbecco’s Modified Eagle Medium (DMEM), Penicillin/Streptomycin, MEM non-essential amino acids 100×, Trypsin-EDTA, L-glutamine, putrescine, spermidine, and spermine. Foetal bovine serum (FBS) was from Gibco Life Technologies, Sao Paolo, Brazil. The protease and phosphatase inhibitors cocktail were from Roche, Indianapolis, IN, USA. RIPA lysis buffer 10× from Millipore, Burlington, VT, USA. The following antibodies were purchased from Abcam Biotechnology, Cambridge, UK: rabbit polyclonal anti-β-actin (ab119716, 1:7500 dilution), goat polyclonal anti-rabbit IgG-HRP (ab205718, 1:10,000 dilution), goat polyclonal anti-AZ1 (ab223481, 1:300), donkey anti-goat IgG H&L-HRP (ab97110, 1:2000). Mouse monoclonal anti-ODC (sc-398116, 1:100 dilution) and goat anti-mouse IgG-HRP (sc-2005, 1:2500 dilution) antibodies were from Santa-Cruz Biotechnology, Heidelberg, Germany.

### 2.2. Experimental Procedures

#### 2.2.1. Cell Culture

The human breast cancer cell lines MCF-7 and MDA-MB-231 (from ECACC) were used in this study. The cells were grown under standard conditions of 37 °C and 5% CO_2_ in T75-cm^2^ flasks containing 15 mL DMEM supplemented with 10% (*v*/*v*) FBS, 1% (*v*/*v*) penicillin/streptomycin, and 1% (*v*/*v*) L-glutamine (for MCF-7 medium was further supplemented with 1% (*v*/*v*) non-essential amino acids). In experiments involving addition of exogenous polyamines to the cells, 1 mM aminoguanidine was also added to the growth media to inhibit amine oxidases present in the complete growth medium.

#### 2.2.2. Growth Determination

Cells were grown in 6 cm plates in duplicate and harvested at 4 h and at every 24 or 48 h up to 312 h depending on the experiment. At the indicated time point, cells were harvested by trypsin-EDTA method, transferred to 15 mL tube, and centrifuged for 5 min at 250 g_av_. The supernatant was discarded, and the cell pellet resuspended in 1 mL of complete growth medium. A total of 100 µL of this suspension was sampled for cell counting by trypan-blue exclusion assay. The remaining 900 µL was centrifuged and the supernatant discarded. The cell pellet was washed twice in 2 mL PBS and lysed with 0.2 M perchloric acid (PCA) on ice for 30 min for polyamine extraction and protein precipitation.

#### 2.2.3. Protein Quantification

Protein was quantified by modification of Lowry et al. [18] method. A bovine serum albumin (BSA) standard curve was prepared in the range 0 to 250 µg/mL in 0.3 M NaOH. Both samples and standards were plated in a 96 well flat-bottom plate and exposed to a basic solution containing Cu^++^ for 10 min prior to addition of 0.13 M Folin-Ciocalteau reagent and incubated in the dark for 30 min. Plate was read using a Tecan Sunrise spectrophotometric microplate reader (Mannedorf, Switzerland). Total protein content was expressed as mg/culture.

#### 2.2.4. Polyamine Quantification

Polyamine content was quantified by LC-MS/MS (liquid chromatography–mass spectrometry) as previously described [19]. Briefly, 200 µL of samples and polyamine standards were placed in reaction tubes, 10 µL of 100 µM of 1,7-diaminoheptane was added to each tube as internal standard, plus 50 µL of 1 g/mL sodium carbonate and 500 µL of freshly prepared 10 mg/mL dansyl chloride in acetone. Tubes were left overnight at 37 °C. The next day, 0.5 mL of toluene was added to extract the dansylated products. The organic phase was evaporated to dryness under N_2_ flow; the aqueous phase was reconstituted in 200 µL methanol and vortex mixed for 10 s. This was then analyzed using liquid chromatography in tandem with mass spectrometry (Thermo Scientific, Hemel Hempstead, UK). Results were expressed in nmol and normalized with total protein content for each sample.

#### 2.2.5. Quantification of Intracellular DFMO

Intracellular DFMO content after cell exposure was quantified by LC-MS/MS using a protocol developed by our group as part of this study.

DFMO stock (10 mM) was prepared in 0.2 M PCA and diluted to 2 mM in PCA. DFMO calibration standards were prepared from this stock using the PCA cell extract from an untreated control to minimize matrix effects during analysis. DFMO standards were prepared in the range 0–20 µM with 50 µL of samples and standards transferred to reaction tubes, followed by the addition of 150 µL of methanol. The tubes were vortex-mixed and centrifuged at 21,000 g_av_ for 5 min. 30 µL of the supernatant from each tube was transferred to respective well of a 96-well plate and analyzed using a Thermo Surveyor–TSQ Quantum system. The column was a Hichrom 3.5µ HIRPB (150 × 2.1 mm) with a mobile phase consisting of 5% water and 95% methanol, both containing 0.1% formic acid, at a flow rate of 0.2 mL/min. The column temperature was maintained at 50 °C and the autosampler tray temperature at 4 °C with an injection volume of 2 µL. The mass spectrometer conditions were set as follows: ionization mode: ESI in positive ion mode, spray voltage: 4000 V, sheath gas: 50, auxiliary: 0, capillary temp: 375 °C, tuned tube lens: tuned value, skimmer offset: −8 V, collision pressure: 1.8 mTorr. Single reaction monitoring was used for the detection of DFMO with the SRM scan filter set at *m*/*z* 183.0–100.1 and a collision energy of 28 V. DFMO eluted at 1.7 min and the total run time for analysis was 2.2 min.

Calibration curves were analyzed using weighted least squares linear regression with a 1/X^2^ weighting. Results were expressed in nmol and normalized with total protein for each sample.

#### 2.2.6. Colony Formation Assay

Cells were seeded in 6 cm dishes for 24 h and treated with drug concentrations for 48 h. Thereafter, cells were harvested and counted. About 500 cells for each treatment per dish were re-seeded in duplicates in a 6 cm dish. Plates were left in a sealed incubator for at least 2 weeks or for eight cycles of generation time. At the completion of incubation, the culture medium in each plate was carefully removed and the cell monolayer rinsed with PBS. The cells were stained and fixed in 2 mL 0.5% (*w*/*v*) crystal violet solution in 50% methanol (*v*/*v* in dH_2_O) for 30 min. The plates were carefully rinsed with tap water and left to dry in normal air at room temperature. Colonies containing 50 cells or more were counted under a Carl Zeiss^TM^ Stemi 2000-C stereo microscope (Jena, Germany) as representative of surviving clonogenic cells. The surviving fraction was determined according to the formula reported by Franken et al. [20].

The surviving fraction is the number of colonies that arise from a single cell after cells were treated with the respective drugs. The surviving fraction was expressed in terms of the plating efficiency for individual treatment taken at 4 h post re-seeding.
$$\mathrm{Surviving}\;\mathrm{fraction}\;\left(\%\right)=\frac{\mathrm{number}\;\mathrm{of}\;\mathrm{colonies}\;\mathrm{formed}\;\mathrm{after}\;\mathrm{treatment}}{\mathrm{number}\;\mathrm{of}\;\mathrm{cells}\;\mathrm{seeded}}\times \;\mathrm{plating}\;\mathrm{efficiency}.$$

#### 2.2.7. Western Blot

Cells were seeded and exposed to the respective drug treatment. After drug exposure, cells were harvested and lysed on ice with ice-cold RIPA lysis buffer containing protease and phosphatase inhibitors. Cell lysates were centrifuged at 17,000× *g* for 15 min at 4 °C. The protein content of the supernatant was quantified by BCA protein assay kit according to the manufacturer’s instructions (Pierce BCA Protein Assay Kit, Thermo Scientific, Rockford, IL, USA). Samples containing equal amount of protein (30 µg) were separated on SDS-PAGE (12% gel) and transferred to a nitrocellulose membrane overnight at 25 V. The membrane was blocked in 5% (*w*/*v*) fat-free milk in PBS-Tween-20 (0.1% (*v*/*v*)) for 1 h, after which membrane was incubated with primary antibodies for 3 h at room temperature. After washing three times for 5 min in PBST, the membrane was incubated with secondary antibodies conjugated HRP for 1 h at room temperature. After incubation, the membrane was washed three times for 5 min in PBST, followed by development using enhanced chemiluminescence substrate (SuperSignal West Dura, Thermo Scientific, Rockford, IL, USA). Blot images were detected by Genoplex Bio Imager (VWR, Radnor, PA, USA). The images were quantified by measuring the intensity of correctly-sized bands using ImageJ software and normalized to the actin band intensity for all experiments.

#### 2.2.8. Polysome Analysis

Cells were seeded and treated with respective drugs based on the experiment. After drug exposure, cells were briefly treated with 100 µg/mL cycloheximide for 15 min, rinsed in cycloheximide containing PBS, and detached by trypsinization. Cells were collected by centrifugation and the pellet was washed twice in ice-cold PBS containing 100 µg/mL cycloheximide, resuspended in 500 µL of complete lysis buffer [25 mM Tris-HCl (pH 7.4), 50 mM KCl, 5 mM MgCl_2_, 2 mM DTT, 100 µg/mL cyclohexamide and 200 µg/mL heparin, 1% triton-100, 1% sodium deoxycholate, and protease inhibitor]. Cells were lysed by pumping the solution up and down a 25-gauge needle three times and left on ice for 20 min while vortexing intermittently. Tubes were centrifuged at 17,000 g_av_ for 15 min at 4 °C. Total protein in the supernatant was estimated on nanodrop at 280 nm. Equal amount of each samples corresponding to 20 OD was carefully layered onto 11 mL 15–50% (*w*/*v*) sucrose gradient containing [25 mM Tris-HCl (pH 7.4), 50 mM KCl, 5 mM MgCl_2_, 2 mM DTT, 100 µg/mL cyclohexamide, and 200 µg/mL heparin] in polycarbonate tubes. The tubes were ultra-centrifuged at 270,900 g_av_ for 2 h 15 min at 4 °C. Following centrifugation, gradients were unloaded from the top by pumping 60% sucrose solution containing 0.05% bromophenol blue to the bottom of each tube. The content of each gradient was measured by UV absorbance using an Optical Unit UV-1 at a wavelength of 254 nm. Results were recorded graphically using a chart recorder. All equipment was from Pharmacia Biotech, Uppsala, Sweden.

### 2.3. Statistical Analysis

Statistical analysis was performed using Graphpad Prism software version 7. Values shown as the mean of all replicates ± S.E.M (standard error of the mean) in which the number of independent experiments was equal or more than three. Values were analyzed by ANOVA with Dunnett’s post-test or by Students’ *t*-test. A *p* value less than 0.05 was considered statistically significant. Where the number of independent experiments were less than three, values shown are mean ± range.

## 3. Results

### 3.1. Polyamine Pathway Inhibition Affects Breast Cancer Subtypes Differentially

Initial studies characterized the growth (as measured by the total cell number) responses of the two breast cancer cell lines to polyamine pathway inhibition. Exposure of both ER+ MCF-7 and the ER- MDA-MB-231 cells to 5 mM DFMO caused almost 60% decrease in intracellular polyamine content after 48 h (Table 1). However, the effect on growth was less significant (Figure 1), with less than 15% decrease in total cell number observed in both cell lines (Table 1). This suggests that the effects of DFMO exposure on cell growth is gradual and develops over time. At longer exposure however, DFMO induced a cytostatic response in the MCF-7 cells but only decreased the growth rate (i.e., the total cell number) in the MDA-MB-231 cells within the period of growth determination. This thus suggests that the ER+ MCF-7 cells, which have higher polyamine content (Table 1), are more sensitive to polyamine pathway inhibition than the ER- MDA-MB-231 cells.

To further determine the responses of the two breast cancer cell subtypes to polyamine pathway inhibition, a survival assay was used to determine the colony formation of the cell lines after exposure to varying DFMO concentrations.

Similar to the effects on cell growth, both cell lines showed differential colony formation ability after exposure to varying DFMO concentrations. In MCF-7, DFMO exposure significantly inhibited colony formation compared to the MDA-MB-231 cells. MCF-7 cells treated with 50 µM (0.05 mM) and 0.5 mM DFMO showed significant inhibition in the number of colonies formed compared to MDA-MB-231 cells, where colony formation was not affected by these DFMO concentrations (Figure 2). This further indicates the greater susceptibility of MCF-7 cells to polyamine pathway inhibition. At 5 mM however, DFMO exposure completely inhibits colony formation in both cell lines.

### 3.2. DFMO Withdrawal Fails to Rescue Cell Growth Inhibition

To further characterize the polyamine and growth responses of the less sensitive ER-MDA-MB-231 cells to polyamine pathway inhibition, the effect of short (48 h) and continuous DFMO exposures were considered and compared to untreated control. In the short exposure condition, DFMO (at 5 mM) was withdrawn from the culture medium after the initial 48 h exposure and the effect of this treatment was compared to cells that were continuously exposed to DFMO and with untreated cells.

DFMO withdrawal failed to rescue the growth and induced cytostatic effects similar to that observed in cells that were continuously exposed to DFMO (Figure 3a). Also, there was no difference in intracellular polyamine content in cells treated with DFMO for short time and those treated for a longer time, although the total polyamine content in both exposures was significantly lower than in the untreated cells (Figure 3b). Western blot analysis of the ODC protein level showed a gradual decrease with increased time in culture but no changes in the AZ (a negative regulator of ODC) protein level at both treatment conditions (Figure 3c,d). This suggests that the DFMO, although only present in the medium for 48 h, had long-term effects on growth and polyamine metabolism in the cell.

Intracellular DFMO analysis showed build-up of DFMO content in cells that were continuously exposed to DFMO (Table 2). In cells treated with DFMO for 48 h, there was a gradual decrease in DFMO content after withdrawal from the culture medium, but residual DFMO remained measurable in the cell (Table 2) and may be responsible for the polyamine and growth responses observed in these cells.

### 3.3. Exogenous Polyamines Reversed Growth Inhibition

To find appropriate conditions to restore intracellular polyamines to the level needed for cell proliferation, cells that had been pre-treated with 5 mM DFMO were further incubated with exogenous polyamines at different concentrations. After 48 and 72 h of polyamine exposure, cell number was determined, and polyamine content measured.

All exogenous polyamines reversed DFMO growth inhibition. In MCF-7 cells, putrescine addition restored growth to 70% of the untreated control after 48 and 72 h exposure (Figure 4a). Similarly, spermidine reversed growth inhibition in a dose-dependent manner to almost 80% of the untreated control at the highest spermidine concentration (Figure 4b). Exogenous spermine (Figure 4c) caused growth recovery similar to that seen with putrescine. Similar results were obtained for MDA-MB-231 cells (Appendix A).

Polyamine analysis, following cell treatment with DFMO and exogenous putrescine, showed a dose-dependent increase in intracellular putrescine content. The increase in putrescine also restored intracellular spermidine content with no changes in the spermine content of the cells (Figure 5a). Similarly, exogenous spermidine following DFMO pre-treatment caused an increase in the spermidine content with no changes in the putrescine and spermine content (Figure 5b). Spermine addition following initial polyamine depletion also caused an increase in spermidine content (Figure 5c). Both putrescine and spermidine additions caused greater increases in intracellular putrescine and spermidine content in MCF-7 cells than in MDA-MB-231 cells (Figure 5a,b and Appendix A).

Spermine addition also caused an increase in the spermine content but no measurable increase in the putrescine content of the cells (Figure 5c). The increase in the spermidine content by exogenous putrescine and spermine with no changes in both putrescine and spermine content by exogenous spermidine suggests that the growth recovery occurring due to polyamines is partly dependent on the spermidine content of the cell.

### 3.4. Exogenous Polyamines Reversed Translation Inhibition

As cell growth is linked to protein synthesis and general translation state of a cell, the effect of this intracellular polyamine modulation on general translation state of the cells was determined and compared with the untreated control cells using polysome distribution analysis. The polysome profile revealed a substantial inhibition of translation in the cells by DFMO exposure as seen with the decrease in heavier polysome peaks (Figure 6a). A similar result was also seen in the MDA-MB-231 cells (Appendix A). The DFMO treatment also caused a decrease in the polysome to monosome (p/m) ratio (Figure 6b) as estimated for the area under curve, an indication of inhibition of translation initiation following polyamine depletion. However, when spermidine was added to the cells for 24 h after the initial DFMO pre-treatment, polysome analysis revealed a substantial increase in the heavy polysomes with a noticeable decrease in the monosome peak (Figure 6c). Estimating the area under curve showed an increase in p/m ratio by spermidine addition compared to DFMO treated cells (Figure 6d), suggesting an increase in translation initiation by exogenous spermidine.

## 4. Discussion

Research relating to polyamines and polyamine metabolism, especially in cancer, has received significant attention in recent decades owing to the important roles of polyamines in cellular and physiological processes. Functions attributed to polyamines range from regulation of gene expression to control of apoptosis, cell cycle progression, regulation of signaling pathways, and cell proliferation [21,22,23,24,25] with these contributing to cell growth, particularly in cancer. While a link between elevated polyamine content and poor prognosis of breast cancer has been established by various studies [14,26], breast cancer patients do not seem to benefit from mono-therapeutic inhibition of the polyamine pathway [27,28,29]. However, in a clinical trial study by O’Shaughnessy, et al., one patient with metastatic breast cancer was found to have stability of liver metastasis for eighteen months on DFMO therapy [28], suggesting that some breast cancer patients might benefit from therapy that targets polyamine metabolism.

The main reason for the general poor efficacy of polyamine pathway inhibitors in cancer treatment is attributed to the compensatory polyamine uptake by the tumour cells from the polyamines present in the diet and polyamines produced by the microbiota in the gut. To counter this, a polyamine blocking therapy that combines both polyamine pathway inhibitor (DFMO) and inhibitor of polyamine transporter (AMXT 1501 dicaprate) has since been developed [30] and is in phase 1 clinical trials for treating patients with advanced solid tumours (ClinicalTrials.gov. Identifier: NCT03536728). This polyamine blocking therapy might thus serves a better therapeutic benefit in breast cancer treatment than the use of monotherapy targeting polyamine pathway inhibition, however, there is a need to determine which breast cancer, based on the hormone receptor status, might benefit most from therapy that depletes intracellular polyamine content.

In this study, the responses of two breast cancer cells of different hormone receptor status to intracellular polyamine modulation was compared. It was shown that the ER+ MCF-7 cells, with higher polyamine content, showed greater growth inhibition to intracellular polyamine depletion than the ER- MDA-MB-231 cells. This greater growth inhibition was further supported by the greater inhibition of colony formation by the different concentrations of DFMO used in the MCF-7 cells. MCF-7 cells also showed higher intracellular polyamine content following exposure to exogenous polyamine than did the MDA-MB-231 cells, suggesting that the ER+ MCF-7 cells are more dependent on polyamines for proliferation and are more susceptible to conditions that limit polyamine content. A study by Kremmer, et al. also suggests MCF-7 cells to be sensitive to the anti-tumour effects of polyamine depletion [31]. Similarly, in the ER+ ZR-75 breast cancer cell line, polyamine depletion by DFMO was found to abrogate the growth effect of oestradiol [32]. These studies indicate greater dependence of hormone receptor positive (HR+) breast cancer cells on polyamines for proliferation, which thus suggests that patients with HR+ breast cancer subtype may benefit from therapies that decrease tumour polyamine content. This potential polyamine limiting therapy in HR-positive breast cancer subtype warrants further investigation that involves the use of other breast cancer cell lines and animal models of the different hormone receptor status.

In addition to the differential hormone receptor status of the two cell lines used in this study, differences in the expression of key proteins involved in cell proliferation and survival, such as p53 and c-Myc, may contribute to the differential growth responses to polyamine modulation. MDA-MB-231 cell expresses mutated TP53 gene, while MCF-7 cell expresses wild-type TP53 gene [33]. Both cell lines also have elevated c-Myc protein, although the mechanism of c-Myc elevation differs. In MCF-7, elevated c-Myc protein is linked to oestrogen/oestrogen receptor stabilization of c-Myc protein half-life [34]. In MDA-MB-231 cells, the c-Myc protein level is associated with decreased Axin1 gene expression [35]. In MCF-7 cell, oestrogen also positively regulates ODC1 gene expression and protein activity [7]. c-Myc on the other hand positively regulates ODC1 gene expression [36,37] and along with spermine synthase promotes cancer cell survival [38]. Thus, differences in gene expression patterns together with the hormone receptor status of the cells may contribute to their differential responses to polyamine pathway inhibition and may help predict the sensitivity of breast cancers to polyamine targeting therapy.

One important observation in this study was that residual DFMO remained measurable in the cell several days after its withdrawal from the culture medium. This finding is particularly interesting and, to the best of our knowledge, this is the first study that has measured intracellular DFMO content after in-vitro exposure. This knowledge of persistent DFMO in cells with accompanied decrease in polyamine content and cell growth will be useful in future cancer therapeutic settings that target polyamine pathway for inhibition.

In determining the requirement of the cells for each polyamine for proliferation, data from this study suggests that the growth effects of the polyamines are partly dependent on the spermidine content of the cell. This observation was supported by the complete restoration of spermidine content by exogeneous putrescine and increase in spermidine by exogenous spermine with little or no change in putrescine and spermine content by exogenous spermidine. This dependence on spermidine may be attributed to its requirement in hypusination/activation of eukaryotic translation initiation/elongation factor 5A (eIF5A), which requires the presence of spermidine for activation. Activation of eIF5A is needed in protein translation, particularly the translation of proteins with polyproline tracts or PPX (where X may be Gly, Trp, Asp, or Asn) [39,40]. These proteins are known to be involved in metabolic processes that promote cell proliferation [41] and hence cancer cell growth. Since breast cancers have elevated polyamines and are more dependent on spermidine for proliferation, therefore, another approach to targeting polyamine biology in breast cancer treatment could be to inhibit the link between spermidine and eIF5A by inhibiting the enzymes (deoxyhypusine synthase and deoxyhypusine hydroxylase) that catalyze the hypusination reaction.

Although a limited model system of each hormone receptor status of the breast cancer subtype was used in this study, the data presented here nevertheless provides strong evidence that breast cancer cells respond to polyamine inhibition differentially and that a non-linear relationship exists between the polyamines in promoting the cancer cell growth. This data also provides the basis for further characterization of more models of breast cancer to polyamine targeting therapy. While higher concentrations of DFMO than clinically achievable in dosing were used in this study, these concentrations were chosen based on prior literature search and previous data from our lab.

In conclusion, we showed that breast cancer cell growth in-vivo is causally linked to their polyamine content and, the sensitivity of the cells to growth inhibition by agent(s) that limit polyamine content is partly dependent on the hormone receptor status of the cells. We also showed that the growth recovery effects of exogenous polyamines were partly dependent on their conversion to spermidine. Polyamine depletion inhibits translation initiation and exogenous spermidine was able to reverse this inhibition in the breast cancer cells. Finally, we suggest that from these results, targeting polyamine metabolisms might be beneficial in ER-positive breast cancers.

## Figures and Tables

**Figure 1 biomolecules-11-00743-f001:**
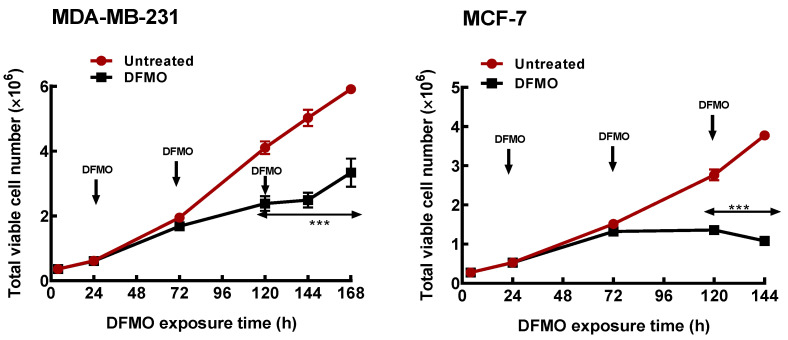
Time course of polyamine depletion on growth response of breast cancer cells. MDA-MB-231 and MCF-7 cells were cultured in growth media treated with DFMO and total cell numbers were determined every 48 h. Values shown are mean ± S.E.M, n = 3–4 with 2 replicates per treatment per independent experiment and analyzed by Students’ *t*-test, *** *p* < 0.001 compared to the untreated control. Arrows pointing down indicate points of DFMO addition and replenishment at 5 mM in the growth media as the media is refreshed.

**Figure 2 biomolecules-11-00743-f002:**
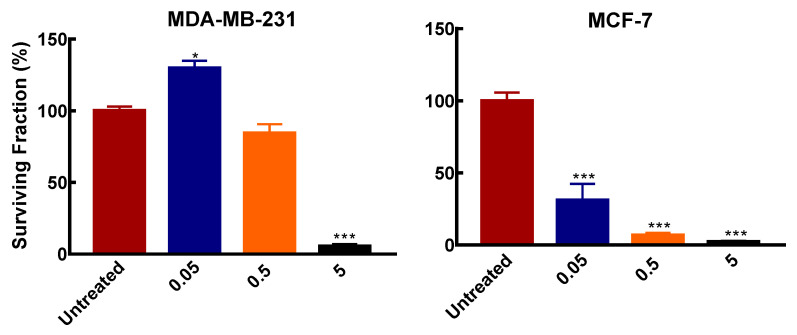
Effects of DFMO exposure on colony formation in breast cancer cells. Cells were seeded in 6 cm dishes at a density of 2.4 × 10^4^/cm^2^. Medium was refreshed after 24 h with varying DFMO concentrations and grown for 48 h. Thereafter cells were reseeded in 6 cm dish at 500 cells/dish without DFMO and incubated for 2 weeks. After incubation, cells were processed as described in Section 2.2.6. Result shown are mean ± S.E.M, (n = 3) with 2 replicates per treatment per independent experiments, * *p* < 0.05, *** *p* < 0.001 compared to the untreated control.

**Figure 3 biomolecules-11-00743-f003:**
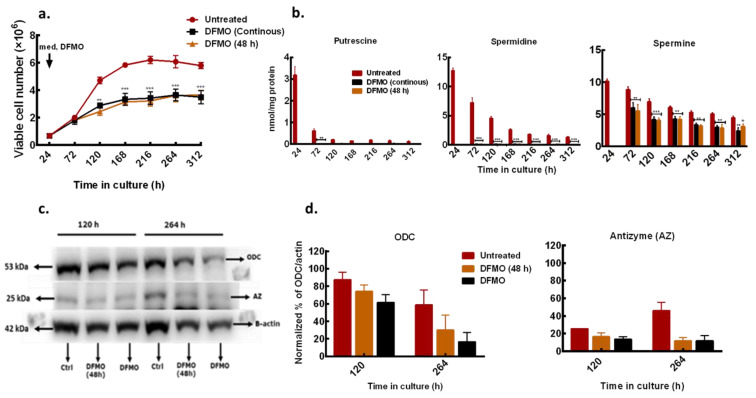
Effects of different DFMO exposure conditions on MDA-MB-231 cells. MDA-MB-231 cells were cultured in growth medium treated with DFMO for 48 h only or continuously every 48 h. Cell numbers were determined every 48 h. (**a**) Time course growth profile of MDA-MB-231 cells treated with different DFMO exposure conditions. (**b**) Total cellular polyamine content. (**c**) ODC and AZ protein level determined in 12% SDS PAGE. β-actin was used as loading control. Ctrl represents untreated control. (**d**) Densitometry of protein bands of two independent experiments from (**c**) were quantified using ImageJ software. Values shown are mean ± S.E.M, n = 3 with 2 replicates per treatment per time point, * *p* < 0.05, ** *p* < 0.01, *** *p* < 0.001 compared to the untreated control.

**Figure 4 biomolecules-11-00743-f004:**
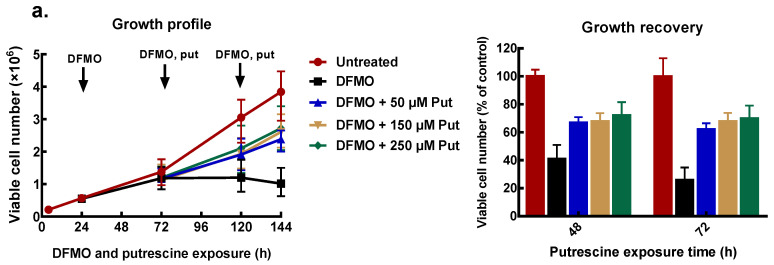
Effect of exogenous polyamines on growth of MCF-7 cells. MCF-7 cells were pre-treated with 5 mM DFMO for 48 h, thereafter cells were treated with different concentrations of polyamines as indicated, following which the cell number was determined. Cell growth profile and percentage growth recoveries by exogenous polyamines: (**a**) putrescine, (**b**) spermidine, and (**c**) spermine. Values shown are mean ± range (n = 2) with 2 replicates per treatment.

**Figure 5 biomolecules-11-00743-f005:**
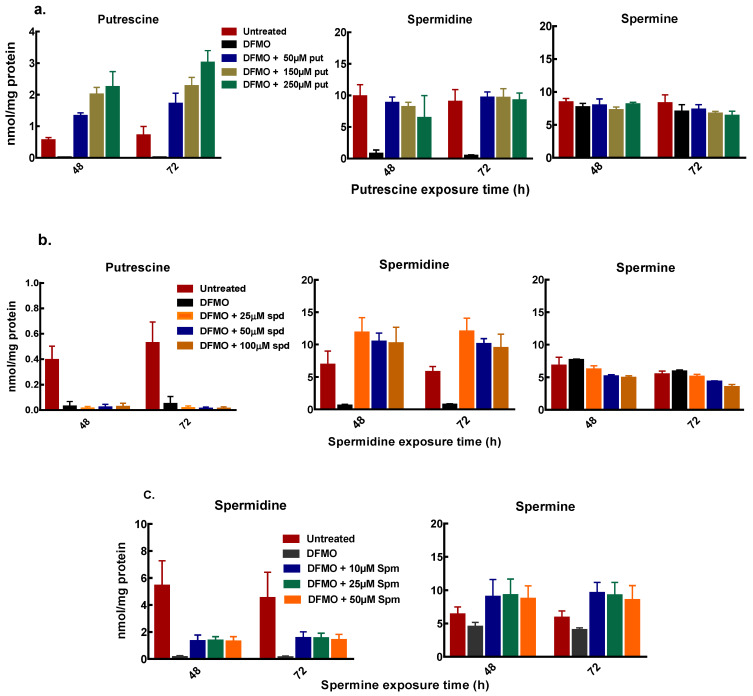
Intracellular polyamine content of MCF-7 cells after exogenous polyamine additions. MCF-7 cells were pre-treated with 5mM DFMO for 48 h, thereafter cells were treated with different concentrations of polyamines as indicated, following which cells were harvested and polyamines quantified. (**a**) polyamine content after exogenous putrescine addition (**b**) polyamine content after exogenous spermidine addition (**c**) polyamine content after exogenous spermine addition. Values shown are mean ± range (n = 2) with 2 replicates per treatment.

**Figure 6 biomolecules-11-00743-f006:**
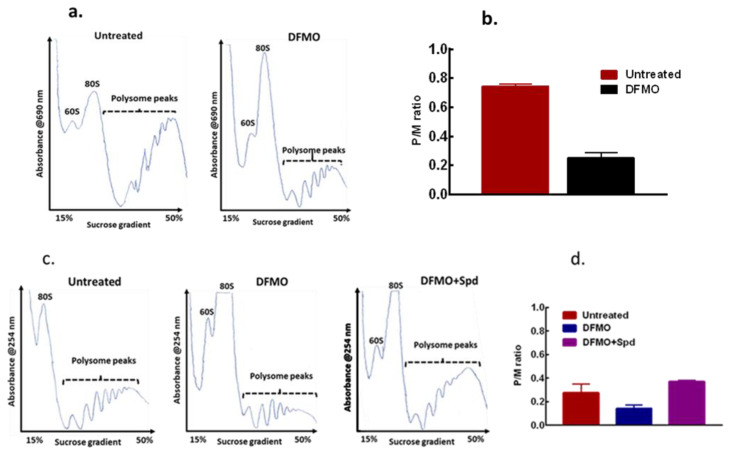
Effects of polyamine modulation on translation initiation of MCF-7 cells. MCF-7 cells were seeded, treated accordingly, and harvested for polysome analysis. (**a**) Representative of polysome profiles from untreated control and 5 mM DFMO treatment (48 h), n = 2. (**b**) Estimation of the polysome to monosome ratio for the respective treatment for the area under curve for (**a**). (**c**) Representative of polysome profiles from untreated control, 5 mM DFMO treatment (72 h) and 5 mM DFMO + 100 µM spermidine (24 h), n = 2. (**d**) Estimation of the polysome to monosome ratio for the respective treatment for the area under curve for (**c**). For DFMO+Spd, the error bar is within the graph.

**Table 1 biomolecules-11-00743-t001:** Effect of DFMO on polyamine content of MCF7 and MDA-MB-231 cells. Polyamine content of 0.2 M perchloric acid cell extract of MDA-MB-231 and MCF-7 cells was analyzed by LC-MS. Values shown are mean ± S.E.M, n = 3−4 with 2 replicates per treatment per independent experiment, * *p* < 0.05 compared to the untreated control, nd means not determined.

DFMO Exposure Time (h)	Treatment	MDA-MB-231			MCF-7		
		Total PA (nmol/mg Protein)	PA Decrease (% of Untreated)	Growth Inhibition (% of Untreated)	Total PA (nmol/mg Protein)	PA Decrease (% of Untreated)	Growth Inhibition (% of Untreated)
0	Untreated	20.7 ± 1.7	0	0	22.7 ± 1.4	0	0
48	Untreated DFMO	18.8 ± 1.3 7.5 ± 0.5 *	60	13	23.3 ± 1.1 9.5 ± 0.4 *	59	12
96	Untreated DFMO	11.6 ± 0.8 5.2 ± 0.1 *	55	42	15.1 ± 0.6 7.8 ± 0.4 *	48	50
120	Untreated DFMO	11.3 ± 1.3 6.2 ± 0.2 *	45	50	14.4 ± 0.9 6.6 ± 0.4 *	54	70
144	Untreated DFMO	7.0 ± 1.2 4.3 ± 0.3 *	39	43	nd	nd	nd

**Table 2 biomolecules-11-00743-t002:** Intracellular DFMO content of MDA-MB-231 cells after withdrawal from growth medium. DFMO content from the 0.2 M perchloric acid cell-soluble fraction was analyzed by LC–MS. Values shown are mean ± range (n = 2) with 2 replicates per treatment per time point.

Time in Culture (h)	DFMO Exposure Time (h)	DFMO Conc. (5 mM)	Intracellular DFMO (nmol/mg protein)
72	48	48	9.9 ± 2.5
120	96	Continuous 48	15.5 ± 2.5 3.3 ± 0.8
168	144	Continuous 48	18.6 ± 1.3 1.4 ± 0.3
216	192	Continuous 48	16.8 ± 1.6 0.6 ± 0.1
264	240	Continuous 48	20.1 ± 1.0 0.3 ± 0.0
312	288	Continuous 48	26.7 ± 6.3 0.1 ± 0.0

## Data Availability

All data used in the preparation of this manuscript is secured and stored at the University of Aberdeen. This data is available for inspection and review at any time.

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
