# Peer review of "Characterising the Response of Human Breast Cancer Cells to Polyamine Modulation"

_biomolecules, 2021, doi:10.3390/biom11050743_

Round 1
Reviewer 1 Report
This paper by Akinyele and Wallace examines the differential response of ER+ (MCF-7) and ER- (MDA-MB-231) breast cancer cells to polyamine modulation, using both short-term and long-term treatment with DFMO to deplete polyamines followed by addition of exogenous putrescine, spermidine or spermine. Parameters measured include cell growth profiles, colony formation assays and polysome profiles to address effects on translation initiation. Importantly, the manuscript describes a novel LC/MS method for measuring intracellular DFMO. Using this method, the authors show that DFMO is maintained at fairly high levels long after it is removed from the culture medium. This observation adds significant new knowledge to the field, and the ability to monitor intracellular DFMO would be potentially useful in a therapeutic setting. However, there are questions regarding data interpretation of the polyamine supplementation experiments, as well as other questions, that should be address as follows:
• Referring to figures 4 and 5 (for MCF-7 cells) and supplemental figures 1 and 2 (for MDA-MB-231 cells), the authors conclude that spermidine controls growth of both cell lines. In fact, spermidine levels are about 5-fold higher in putrescine-treated MCF-7 cells compared to spermine-treated cells (fig 5), yet the growth recovery is very similar (fig 4, about 70%). This does not seem to be consistent with the idea that spermidine alone is controlling growth. In addition, all 3 polyamines are able to recover growth equally in the MDA-MB-231 cells (supplemental figures 1 and 2). This again does not appear to be consistent with the conclusion that spermidine controls growth. Additional explanation of these results is needed.
• The polysome profiles in Figure 6 compare DFMO to rapamycin treatment. What is the effect of rapamycin on polyamine levels? It is known to inhibit ODC synthesis.
• AZ levels are quite inconsistent in the long-term DFMO treated cells. This should be addressed.
• In the colony formation assays shown in Fig 2, the surviving fraction appears to be very low even in untreated cells. Is this correct?
• MDA-MB-231 cells are a triple negative breast cancer line (ER, PR and HER2 negative), and also express a mutated p53. Thus, the difference in results seen may not be due to estrogen receptor status alone. This warrants some discussion.
Author Response
Thank you for your careful reading of our manuscript and for the helpful comments. We have addressed these as indicated below.
Point 1
While we understand that each polyamine was able to reverse growth inhibition in both cells, our argument is that the restoration and/or increase in spermidine content by exogenous putrescine and spermine respectively is partly responsible for their growth effects.
Point 2
The polysome profile for the rapamycin-treated cells was added as a positive control to the polysome profile in DFMO treated cells. However, as this appears to complicate interpretation of the data this has now been removed.
Point 3
We are aware that the AZ data is inconsistent with the ODC protein level in the long-term DFMO treated cells, this data was added as an indication that we made attempt to look at the AZ protein level in this experiment.
Point 4
The surviving fraction of the untreated cells was not low, it was due to the way the data was initially presented. We have now represented the data as a percentage of the untreated cells.
Point 5
A discussion on the p53 status of the cells has now been added.
Please see the attached manuscript

Reviewer 2 Report
Overall comments:
This is a nicely presented and designed series of experiments characterizing the effects of DFMO on cell growth/viability, intracellular polyamine content, and protein translation. The data are generally well presented, though there are several typo’s throughout, which I have attempted to correct as listed below. A careful re-read/edit will be necessary prior to the potential for publication.
Limitations include that only two cell lines were used for all of the experiments, and the authors made some conclusions based on breast cancer cell “subtype” (see comments below, but I believe the reference is to hormone receptor status). However, in only looking at one cell type in each “subtypbe” it is very difficult to attribute differences in DFMO responses based on each cell line’s hormone receptor status, i.e. it may be related, but there may be MYC gene mutations/amplifications that may have a much more important role in predicting response to DFMO, as protein translational output is very dependent on MYC in many cancer types. And of course, there are many other cell-specific factors that may be leading to these differences, and this really doesn’t become apparent until multiple cell lines are tested from the same hormone receptor subtype. For the purpose of this manuscript, it may not be necessary to perform all of the experiments on additional cell lines, but may be worth further characterizing the intracellular polyamine effects, for example, in additional cell lines.
It may be worth commenting at the end of the discussion on limitations of this study, including limited model systems, much higher than clinically achievable dosing, etc.
Minor questions/comments:
For Figure 1: when the arrow indicates additional DFMO treatment, it is important to clarify that you are in fact not adding more to the culture medium present (i.e. changing the concentration), but instead replacing the DFMO at the same concentration in fresh culture medium.
DFMO Withdrawal Fails to Rescue Cell Growth Inhibition: At the end of this section, and before Figure 3, the authors seem to link the decrease in ODC levels decreasing with long term effects on growth and polyamine metabolism in the cell. The DFMO initial exposure and then removal clearly had effects on growth and total intracellular polyamine content. However, the data presented are not able to link changes with ODC1 protein levels with either growth or polyamine content. If anything, the western blot demonstrates that there are minimal changes in ODC content after 120 hours of DFMO exposure but a more significant decrease seen at 264 hours of continuous exposure. ODC1 is known to be a very tightly regulated protein (many layers of regulation at both transcription and translational), but it seems that it takes a long exposure to DFMO to truly reduce the levels. Were these Western blots performed with the other cell line used in the other experiments? If not, it would be worth looking at ODC1 levels in that cell line at different timing of DFMO exposure.
Specific comments/edits/suggestions by line:
33: “fifth causes” I believe you mean “fifth leading cause of cancer-related death”
36: “event in” maybe better as “driver of”
41: “all human cancers”. Is this true of really every one? It may be safer to say something like “the vast majority of” or something along those lines
48: “prognosis of breast cancer” to “prognosis in breast cancer”
48: “tumour size, histological grade” to “tumour size and histological grade”
50: “and the disease” instead to “and disease”
52: specify what type of “subtypes” you mean, i.e. differences in histology, hormone receptor status, genetic mutational status, etc?
52: “similar response” Do you mean response to polyamine depletion? Or response to what? Should clarify.
55: “polyamine content and determined the” should be “polyamine content and to determine the” to maintain grammatical parallel structure
55: subtype to subtypes, and again here, should specify what differences you are referring to
58: “of key enzyme” to “of a key enzyme”
58: “of polyamine biosynthesis” to “in polyamine biosynthesis”
60: This sentence is a little confusing. Maybe “We also determined the requirement of each polyamine of both breast cancer cell line for growth…”
62: “global translation state” do you mean protein translational state? If so, this may be clearer to state as such
146: “Colony containing 50 cells, or more…” to “Colonies containing 50 cells or more”
153-154: equation explained here does not visually translate well in the version I got, but hopefully that is a formatting thing easily fixed
203: remove “in both cell lines”, this is implied from how the sentence is started
203: when you say “growth” it may be better to describe this as “growth kinetics” or something along those lines, as these are not physical tumors growing in size, but instead absolute numbers of cells present
205: “This suggest that” should be “This suggests that”
207: “only growth inhibition” should instead be something about growth rate inhibition instead, as you didn’t measure cell or colony size, but instead measured absolute cell number
214: “cells number” should be “absolute cell counts” or something similar
216: Was DFMO added in an absolute sense, or replaced with culture media replaced. I assume since the concentration used was listed as 5 mM, it wasn’t drug being truly added by instead replaced? Please clarify
228: add comma “compared to MDA-MB-231 cells, where”
243: what is the definition here of prolonged exposure?
245: “the effect of this was” to “the effect of this treatment was”
246: remove “for a longer time and”
246: “(prolonged) exposed to DFMO with untreated cells” to “(prolonged) exposed to DFMO and with untreated cells”
251: remove extra space between “in the”
258: “Cells number” should be “Cell number”
259: Figure 3 legend part (b) would be more clear if it read “Total cellular polyamine content”. Not sure you need to reference
275: “After 48 and 72h polyamine” to “After 48 and 72h of polyamine”
280: “highest concentration of spermidine used” to “highest spermidine concentration”
281: Remove “While”
Figure 4c: In the legend, capitalize “Spm” as you did with the two other graphs of Put and Spd
291: “Polyamine analysis shows…” This should explain that you looked at polyamine analysis was performed after DFMO treatment. The way this currently reads, it is not clear what treatment parameters were used in general in these experiments.
291: “in putrescine content” should specify “in intracellular putrescine content”
318: add commas and change to “mTOR, and hence protein translation processes, and…”
346: “from monotherapy that inhibits polyamine pathway” to “mono-therapeutic inhibition of the polyamine pathway”
348: “Although” to “However”
350: “on DFMO administration” should be “on DFMO therapy”. (not really just on administration, but instead clinically best referred to as being on DFMO therapy.
351: “target polyamine” should be “targets polyamine”
360-362: Ned to rewrite this sentence to truly talk about what “subtype” you really mean. Here it says “cancer subtype”, but I assume you mean breast cancer, and by subtype meaning hormone receptor status.
370: “ER+ MCF-7 cell is” should be “ER+ MCF-7 cells are”
372: remove “that” from “showed that MCF-7 cells”
373: should be consistent with ER-positive vs. ER+. Both have been used here.
376: “cells on polyamine for” to “cells on polyamines for”
377: “breast cancer subtype might” should be “breast cancer may”
378: “therapy that decrease to “therapies that decrease”
380: “cell and animal models” should be “cell lines and animal models”
382: “in the cell several days after withdrawal” to “in the cells several days after its withdrawal”
382: “This is particularly” to “This finding is particularly”
383: add commas as follows “and, to the best of our knowledge, this”
384: “cell exposure” should be changed to “in vitro exposure”
385: “might guide in” should be “may guide”
386: Clincial doses (recommended phase 2 dosing, etc) has already been established in adults and peds for DFMO. This sentence should be changed to discuss the use fo DFMO in future cancer therapy, not the dosing of it. And of note, anyting related to dose is very hard to draw any conclusions from in this study, as “tool compound doses” of DFMO were used (5 mM) as opposed to the 100-300 microM that is attained in humans after dosing of up to 9 g/m2. So dosing conclusions really cannot be made from these studies.
392: “This dependent” should be “This dependence”
393: add to “… (eIF5A).” to say “…(eIF5A), which requires the presence of spermine for activation”
402: “cell growth” to “cell growth in vivo”
Author Response
Thank you for your careful reading of our manuscript and for the helpful comments. We have addressed these as indicated below.
Overall comments
Point 1
We have added a discussion on the c-Myc status of the cells.
Point 2
Comment on the limitations of this study has now been added as suggested.
Minor questions/comments
Point 1
This has now been clarified to indicate that the DFMO concentration was unchanged and that same concentration was added each time along with the growth media as the media is refreshed.
Point 2
Due to the limitation of time, we could not carry out these experiments in the other cell line, but as suggested this would be considered in future studies.
Specific comments/edits/suggestions by line:
All these have been corrected.
Please find the manuscript attached for the corrections

Round 2
Reviewer 1 Report
The authors have answered all concerns adequately and the manuscript is now acceptable for publication